# True Prevalence of Unforeseen N2 Disease in NSCLC: A Systematic Review + Meta-Analysis

**DOI:** 10.3390/cancers15133475

**Published:** 2023-07-03

**Authors:** Wing Kea Hui, Zohra Charaf, Jeroen M. H. Hendriks, Paul E. Van Schil

**Affiliations:** 1Department of Thoracic and Vascular Surgery, University Hospital Antwerp, Drie Eikenstraat 655, 2650 Edegem, Belgium; jeroen.hendriks@uza.be; 2Department of Cardiothoracic Surgery, University Hospital Brussels, Laarbeeklaan 101, 1090 Jette, Belgium; 3ASTARC (Antwerp Surgical Training, Anatomy and Research Centre), University Hospital Antwerp, Drie Eikenstraat 655, 2650 Edegem, Belgium

**Keywords:** non-small cell lung cancer, nodal disease, surgery

## Abstract

**Simple Summary:**

Preoperative mediastinal staging plays a crucial role in determining the appropriate treatment strategy for patients with stage IIIA-N2 disease, but an optimal treatment strategy has yet to be established. Invasive mediastinal staging is indicated in approximately 30% of patients with suspected NSCLC. In general, if proven N2 disease is present, induction therapy is prioritized in order to downstage and achieve a better prognosis. In the absence of N2 disease, surgical resection with mediastinal lymphadenectomy is the most appropriate treatment option. Nevertheless, unforeseen N2 (uN2) disease, also called unexpected or surprise N2, can still be found during or after surgery despite current preoperative mediastinal staging showing N0 or N1 disease. As preoperative mediastinal staging improved over time, the prevalence of uN2 changed. The aim of this study is to determine the prevalence of true uN2 disease and its characteristics. A secondary objective is to identify its significance for long-term outcomes.

**Abstract:**

Patients with unforeseen N2 (uN2) disease are traditionally considered to have an unfavorable prognosis. As preoperative and intraoperative mediastinal staging improved over time, the prevalence of uN2 changed. In this review, the current evidence on uN2 disease and its prevalence will be evaluated. A systematic literature search was performed to identify all studies or completed, published trials that included uN2 disease until 6 April 2023, without language restrictions. The Newcastle-Ottawa Scale (NOS) was used to score the included papers. A total of 512 articles were initially identified, of which a total of 22 studies met the predefined inclusion criteria. Despite adequate mediastinal staging, the pooled prevalence of true unforeseen pN2 (9387 patients) was 7.97% (95% CI 6.67–9.27%), with a pooled OS after five years (892 patients) of 44% (95% CI 31–58%). Substantial heterogeneity regarding the characteristics of uN2 disease limited our meta-analysis considerably. However, it seems patients with uN2 disease represent a subcategory with a similar prognosis to stage IIb if complete surgical resection can be achieved, and the contribution of adjuvant therapy is to be further explored.

## 1. Introduction

Lung cancer remains one of the leading causes of cancer-related deaths worldwide [1]. Preoperative mediastinal staging plays a crucial role in determining the appropriate treatment strategy for patients with stage IIIA-N2 disease, but an optimal treatment strategy has yet to be established. Since the turn of the century, we have witnessed an increase in the accuracy of mediastinal nodal assessment as the use of computed tomography (CT) in combination with positron emission tomographic (PET) scanning followed by minimally invasive techniques (endobrochial ultrasonogaphy (EBUS), endoscopic ultrasonography (EUS), and mediastinoscopy) has become routine in the work-up of patients suspected of having lung cancer. The current European Society of Thoracic Surgeons (ESTS) guidelines are illustrated in Figure 1 [2]. Invasive mediastinal staging is indicated in approximately 30% of patients with suspected NSCLC. In general, if proven N2 disease is present, induction therapy is prioritized in order to downstage and achieve a better prognosis. In the absence of N2 disease, surgical resection with mediastinal lymphadenectomy is the most appropriate treatment option. Nevertheless, unforeseen N2 (uN2) disease, also called unexpected or surprise N2, can still be found during or after surgery despite current preoperative mediastinal staging showing N0 or N1 disease. Initially reported prevalence rates were high, up to 25%, with low survival rates as earlier studies reporting on uN2 did not comprise the current gold standard for initial non-invasive mediastinal staging, CT scanning in combination with PET or integrated PET-CT [3]. Incorporating PET scanning has effectively lowered the rate of uN2 disease [4,5]. It is presumed that in contemporary patients with uN2 disease, only minimal node metastasis is found, which is associated with reasonable long-term survival rates if complete resection can be obtained. As preoperative mediastinal staging improved over time, the prevalence of uN2 changed. The aim of this study is to determine the prevalence of true uN2 disease and its characteristics. A secondary objective is to identify its significance for long-term outcomes. The present study was performed to assess the current evidence on uN2 disease as these patients are traditionally considered to have an unfavorable prognosis.

## 2. Materials and Methods

### 2.1. Search Strategy

We sought to identify all studies or completed, published trials that include uN2 disease. A systematic literature search was performed using MEDLINE (via PubMed). To maximize the sensitivity of the search strategy and identify all relevant studies, the following search terms were used in the title or abstract field as well as their related thesaurus terms: non-small cell lung carcinoma, thoracic surgery, metastatic lymph node, and uN2. (The full search strategy can be found in Appendix B). Reference lists of all relevant studies were checked to identify additional relevant articles. No restrictions were placed on the language or date of publication. The last search was performed on April 6, 2023. Inclusion criteria were surgical patients with cN0-1pN2 NSCLC (uN2) with no induction therapy and complete resection (R0) who underwent preoperative imaging by CT and PET scan. To avoid bias by ignoring N2 nodal involvement (after suspicious nodes on imaging studies, lack of invasive preoperative staging, and intraoperative staging), nodal assessment had to be based on preoperative invasive mediastinal staging if imaging tests showed suspicious mediastinal nodes, and intraoperative lymph node sampling or dissection. When multiple publications from the same institution were found, we included the paper that was most complete or suitable for our study’s objective in order to prevent double counting of patient cohorts. Eligible sub-cohorts within studies were also included in the analysis if inclusion criteria were met. We excluded cohorts focused on primary underlying diseases other than non-small lung cancer. Additional exclusion criteria were guidelines, conference abstracts repeating data from included studies, editorial reviews, and studies reporting insufficient data regarding uN2 disease. Two investigators (WKH and ZC) independently evaluated the titles and abstracts of all papers identified by the aforementioned search strategy, followed by obtaining full-text articles for further assessment. Discordances were resolved by consulting the senior author (PVS). The following data were extracted from each included paper: first author, publication year, study design, number of patients in the full reported cohort and with uN2 disease, patient characteristics (age and sex), indication and method of preoperative mediastinal staging, operation technique, and long-term outcomes.

### 2.2. Statistical Methods

RevMan software Version 5.4.1 for Windows (the Cochrane Collaboration; Copenhagen, Denmark) was used to perform all statistical analyses. Extracted data were pooled using the inverse-variance method with either a fixed-effects or random-effects model, depending on the degree of heterogeneity. Between-study heterogeneity was assessed using Higgins’ I^2^ statistics. A study analysis with an I^2^ value of >50% was considered to have a high degree of heterogeneity. If heterogeneity was present, the random-effects model was applied, and subgroup analysis was performed.

### 2.3. Quality Assessment

The Newcastle-Ottawa Scale (NOS) was used to appraise the quality of the included studies. The quality of the study was defined on a scale of 0 to 9. A high-quality study was defined as one with a score ≥ 6 and deemed low risk for bias [6]. The assessment was performed independently by two investigators (WKH and ZC). If necessary, the senior author was consulted to settle disagreements.

## 3. Results

### 3.1. Search Results

Our search strategy identified 5866 papers for consideration. Figure 2 shows a study flow diagram according to PRISMA guidelines 2020 [7]. Of these, 5349 were excluded based on review of the abstracts or following identification as duplicates. Of the 512 publications that remained, full manuscripts were obtained and evaluated. After the exclusion of 492 studies, twenty-four were deemed eligible for inclusion, and a further two were excluded for repeat data (Excluded reports are listed in Appendix A). Ultimately, twenty-two studies were included in this review. Data on the prevalence of uN2 and subcategories were extracted from twenty publications (Table 1 and Table 2), and analysis of survival outcomes was based on nine publications (Table 3). Two papers did not allow inclusion for prevalence analysis of uN2 due to a lack of data but were found to be appropriate for inclusion for survival analysis [8,9]. One study reported long-term oncological outcomes but was not included for analysis as complete resection in the full cohort was not obtained [10]. All studies were retrospective, except for one [11]. All included studies were labeled as having a low risk of bias (≥6). The most common quality defect was failing to report the adequacy of follow-up. Intraoperative lymph node assessment is based on the recommended definitions in the ESTS guidelines published in 2006 [12]. We describe systemic lymph node dissection (SLND) as the complete mediastinal lymphadenectomy of all mediastinal tissues in the known anatomical stations that contain lymph node tissue and the removal of all regional hilar nodes based on the lobe removed. Mediastinal lymph node sampling (MLNS) was defined as intraoperative sampling and pathologic evaluation of suspected LN stations from as many mediastinal stations as possible, with emphasis on those proximal to the primary tumor but always including the subcarinal station. Lobe-specific lymph node dissection (LS-LND) was defined as the dissection of mediastinal lymph nodes according to the location of the primary tumor and did not necessarily include the subcarinal station.

### 3.2. True Prevalence of Unforeseen N2 (uN2)

The pooled prevalence of true unforeseen pN2 derived from the selection of studies was 7.97% (95% CI 6.67–9.27%) (*n* = 9387) based on nineteen studies depicted in the forest plot (Figure 3) and funnel plot (Figure 4). Subgroup analysis was performed based on invasive mediastinal staging, type of surgery, intraoperative lymph node assessment, and period of patient accrual. (Appendix C). Table 1 provides an overview of the characteristics of sixteen studies reporting the prevalence of uN2 NSCLC. A total of 7625 patients with cT1-4N0-1, based on CT and PET imaging followed by invasive mediastinal staging methods, underwent surgical resection with curative intent. As all patients were staged with CT and PET imaging, each patient’s accrual took place after 2000, with the majority in the last half of the first decade as PET scanning became more widely available in this period. No trends in the prevalence of uN2 were seen over time. The majority of the included studies reported details on their indication to perform invasive mediastinal staging by either E(BUS) or mediastinoscopy. Three hundred eighty-two patients underwent subsequent invasive mediastinal staging through E(B)US and mediastinoscopy if indicated. Both Citak [13] and Obiols [14] reported minimally invasive video-assisted mediastinoscopy as a routine but performed an extended mediastinoscopy in case of subaortic or paraaortic nodal stations in left-sided tumors. A statistically significant subgroup effect according to period of patient accrual (*p* < 0.00001) and surgery type (*p* < 0.00001) is suggested. However, we observed an uneven contribution of data among the subgroups and substantial remaining heterogeneity between the studies within each of these subgroups, meaning that the analysis is unlikely to produce useful findings. Univariate analysis of the included studies showed that left-sided tumors, adenocarcinoma, vascular invasion, and pleural invasion [15], T stage (cT2a, pT1b), large cell carcinoma, pN1 involvement [16], size and centrally located tumors [16,17], and high SUVmax [15,16,17] were independently significantly associated with the presence of uN2. In multivariate analysis, a significantly higher risk of uN2 was found in central tumors according to Yoon (*p* < 0.001) [17] and Fiorelli (*p* = 0.003) [16], adenocarcinoma and vascular invasion as reported by Bille (*p* < 0.001) [15], T stage (cT2a, pT2a) (*p* < 0.0001), pN1 involvement (*p* = 0.004), and SUVmax > 4.0 (*p* = 0.007) by Fiorelli [16]. Ghaly et al. [18] found that SUVmax > 3.3 was multivariably associated with a higher risk of nodal upstaging (pN1-2) (*p* = 0.016), while Boada [19] could only identify tumor centrality and male sex as independent risk factors for upstaging to pN1, but no significance was reached in nodal upstaging to pN2. No stratified outcomes for the prevalence of uN2 were available for these variables.

**Table 1 cancers-15-03475-t001:** Study characteristics and results reporting on the prevalence of uN2 disease. ^a^ starting from 2011; ^b^ excluding lymph node station 6/9. Abbreviations: LN: lymph node; FDG fluorodeoxyglucose; CM: cervical mediastinoscopy; VATS: video-assisted thoracoscopy; SLND: systemic lymph node dissection; SUVmax: maximum standardized uptake value; EBUS: endobronchial ultrasonography; LS-LND: lobe-specific lymph node dissection; MLNS: mediastinal lymph node sampling; ns: not specified; na: not applicable; RATS: robot-assisted thoracoscopy.

Author, Year and Journal of Publication	Origin and Time Span	Preoperative Invasive Mediastinal Staging		Type of Surgery	Intraoperative Mediastinal Staging	Full Cohort	Unforeseen N2	
		Indications	Method			N	N	%
Amer et al., 2011. Eur J Cardiothorac [20]	UK, 07–11	Single mediastinal LN > 15 mm (short axis) or low-to-moderate FDG-avidity	CM	Open + VATS	SLND	86	8	9.3
Bille et al., 2017. Eur J Cardiothorac Surg [15]	USA, 00–12	LN ≥ 10 mm (short axis), SUVmax ≥ 2.5, exclusion if suspicious N2	EBUS + CM	Open + VATS	SLND	1667	146	8.8
Boada et al., 2019. Lung Cancer [19]	Spain, 11–17	FDG-avid or LN > 10 mm (short axis), central tumors, T > 3 cm	EBUS + CM	Open + VATS	SLND	323	20	6.2
Citak et al., 2018. Zentralbl Chir [13]	Turkey, 10–17	LN ≥ 10 mm (short axis), SUVmax ≥ 3, central or tumor > 3 cm	CM (+extended CM if indicated)	Open	SLND	1038	133	12.8
Fiorelli et al., 2015. Thorac Cardiovasc Surg [16]	Italy, 06–12	LN ≥ 10 mm (short axis) or SUVmax > 2.5, if present exclusion	CM	Open	SLND or MLNS	901	108	12.0
Ghaly et al., 2017. Ann Thorac Surg [18]	USA, 00–15	Exclusion if LN ≥ 10 mm (short axis), SUVmax ≥ 2.5, centrally located or tumor > 2 cm	na	Open + RATS + VATS	SLND	449	23	5.1
Kamigaichi et al., 2023. Eur J Cardiothorac Surg [21]	Japan, 06–14	Mediastinal LN > 10 mm (short axis) or FDG-avid LN	EBUS	ns	LS-LND	438	30	6.9
Kamigaichi et al., 2023. Eur J Cardiothorac Surg [21]	Japan, 06–14	Mediastinal LN > 10 mm (short axis) or FDG-avid LN	EBUS	ns	SLND	438	38	8.7
Kim et al., 2010. J Thorac Cardiovasc Surg [10]	South Korea, 04–08	Not specified	CM	VATS	SLND	547	40	7.3
Kirmani et al., 2018. J Thorac Dis [22]	UK, 06–10	LN > 10 mm (short axis), SUVmax > 2	E(BUS) + CM	ns	MLNS	312	28	9.0
Lucena et al., 2023. Respir Med [11]	Spain, 15–19	LN ≥ 10 mm (short axis) or FDG-avid LN, centrally located or tumor > 3 cm	EBUS + CM	ns	SLND	46	2	4.3
Merritt et al., 2013. Ann Thorac Surg [23]	USA, 08–12	LN > 10 mm (short axis) or SUVmax > 2.5, if present exclusion	CM	Open + VATS	SLND or MLNS	129	6	4.7
Mun et al., 2020. Eur J Cardiothorac Surg [24]	Japan, 08–16	Suspicious N1–2	EBUS (if technically not possible: CM)	VATS	LS-LND	660	36	5.5
Nguyen et al., 2019. Eur J Cardiothorac Surg [25]	USA, 04–13	Not specified	CM (+EBUS ^a^)	RATS	SLND ^b^	71	2	2.8
Obiols et al., 2014. Ann Thorac Surg [14]	Spain, 04–10	Central tumors, tumors with low FDG-uptake, LN > 16 mm or FDG-avid N1	CM (+extended CM if indicated)	Open + VATS	SLND	540	30	5.6
Shingyoji et al., 2014. Ann Thorac Surg [26]	Japan, 08–13	FDG-avid or LN > 10 mm (short axis), central tumors or T > 2 cm.	EBUS	Open + VATS	SLND	106	13	12.3
Visser et al., 2021. Lung Cancer [27]	Netherlands, 07–19	Mediastinal LN ≥ 10 mm (short axis) or FDG-avid LN	CM and/or EBUS	ns	SLND	418	44	10.5
Yazgan et al., 2021. Acta Chir Belg [28]	Turkey, 07–20	Not specified	EBUS and/or CM	Open + VATS	ns	840	69	8.2
Yoon et al., 2019. Korean J Thorac Cardiovasc Surg [17]	South Korea, 05–14	Multiple N1 or N2 FDG-avid LN	E(BUS) + CM	ns	SLND	166	22	13.3
Zirafa et al., 2018. Surg Endosc [29]	Italy, 13–16	LN ≥ 10 mm (short axis), SUVmax ≥ 3	EBUS + CM	Open + RATS	SLND	212	13	6.1

### 3.3. Subcategories of Mediastinal Nodal Metastasis

An initial nine papers contained data on subcategories of uN2 based on the number of involved lymph node metastases: single vs multiple mediastinal station involvement and skip metastasis. We included four papers [14,20,28,30] in which data could be extracted based on the subdivision of lymph node involvement proposed by the eighth edition of the TNM Classification for Lung Cancer [31] (Table 2). Single-station metastasis (N2a2) is defined as a single positive N2 node with N1 involvement, and N2b is defined as multiple N2 nodal station metastases regardless of N1 involvement. Skip metastasis (N2a1) was defined as a single positive N2 node in the absence of positive N1 nodes. However, no statistical analysis was performed for these subcategories due to insufficient data; therefore, we were not able to comment on this subdivision of nodal involvement.

**Table 2 cancers-15-03475-t002:** Overview of studies reporting uN2 subcategories and their prevalence. Abbreviations: LN: lymph node; FDG: fluorodeoxyglucose; CM: cervical mediastinoscopy; VATS: video-assisted thoracoscopy; SLND: systemic lymph node dissection; ns: not specified; nr: not reported.

Author, year and Journal of Publication	Origin and Time Span	Preoperative Invasive Mediastinal Staging		Type of Surgery	Intraoperative Mediastinal Staging	Full Cohort	cN-Description (uN2)		Unforeseen N2	Single (N2a2)		Multiple (N2b)		Skip (N2a1)	
		Indications	Method			N	cN0	cN1	N	N	%	N	%	N	%
Amer et al., 2011. Eur J Cardiothorac [20]	UK, 07–11	Single mediastinal LN > 15 mm (short axis) or low-to-moderate FDG-avidity	CM	Open + VATS	SLND	86	7	1	8	2	2.3	4	4.7	2	2.3
Obiols et al., 2014. Ann Thorac Surg [14]	Spain, 04–10	Central tumors, tumors with low FDG-uptake, LN > 16 mm or FDG-avid N1	CM (+extended CM if indicated)	Open + VATS	SLND	540			30	9	1.7	5	0.9	16	3.0
Park et al., 2019. J Thorac Oncol [30]	South Korea, 04–14	ns	EBUS + CM	ns	Ns	nr			495	257		107		131	
Yazgan et al., 2021. Acta Chir Belg [28]	Turkey, 07–20	ns	EBUS or/and CM	Open + VATS	Ns	840			69	38	4.5	22	2.6	31	3.7

### 3.4. Overall Survival (OS) Outcomes of uN2

Table 3 provides an overview of the characteristics of nine studies reporting on the long-term outcomes of patients with uN2 disease with complete resection, in which OS outcomes for 3y, 5y and 10y were reported in one, nine and one study, respectively. As depicted in the forest plot (Figure 5), the pooled OS at 5 years was 44% (95% CI 31–58%) (*n* = 892). When available studies for OS after 5 years were compared based on geography (Asia (*n*= 597), USA (*n* = 241), and Europe (*n* = 54)), no statistical difference could be found, suggesting that possible biologic differences do not affect the 5-year OS (*p* = 0.58). However, because of the uneven contribution of the data from the subgroups and the substantial remaining heterogeneity between the studies within each of these subgroups, this analysis may not be able to detect subgroup differences. Other subgroup analyses were not performed as there was insufficient data on meaningful characteristics available. Multivariate analysis by Kim et al. [8] identified age (*p* = 0.014) and peripheral vascular disease (*p* = 0.038) as factors that are associated with worse survival in patients with uN2. Mun et al. [24] reported that single-station N2 disease was a prognostic factor for OS (*p* = 0.023) and pathological T-stage for both OS (*p* < 0.001) and recurrence-free survival (*p* = 0.034). Adjuvant chemotherapy significantly increased OS compared with observation alone (*p* = 0.005) in the paper published by Nakagawa [9]. No stratified outcomes for OS rates for uN2 were possible.

**Table 3 cancers-15-03475-t003:** Survival outcomes of the uN2 disease. Abbreviations: y: years; ns: not specified; VATS: video-assisted thoracoscopy; SLND: systemic lymph node dissection; MLNS: mediastinal lymph node sampling; LS-LND: lobe-specific lymph node dissection.

Author, Year and Journal of Publication	Origin and Time Span	N	Mean Age (y ± SD)	Type of Surgery	Intraoperative Mediastinal Staging (%)	Adjuvant Chemotherapy (%)	Adjuvant Radiotherapy (%)	Adjuvant Chemo-Radiotherapy (%)	Overall Survival (%)				DFS (%)			Remarks
									1	3	5	10	1	3	5	
Bille et al., 2017. Eur J Cardiothorac Surg [15]	USA, 00–12	140		Open + VATS	**SLND**						36.4					Patients underwent lobectomy, segmentectomy or wedge resection.
Citak et al., 2018. Zentralbl Chir [13]	Turkey, 10–17	133	58.6 ± 8.1	Open	SLND						26.4					
Kim et al., 2018. J Thorac Dis [8]	USA, 00–11	101	65 ± 11		ns	84/98 (85.7)	84/97 (86.6)				48	24				10% underwent pneumonectomy.
Kirmani et al., 2018. J Thorac Dis [22]	UK, 06–10	28	66.5		MLNS						35.8					Patients underwent lobectomy, segmentectomy or pneumonectomy.
Mun et al., 2020. Eur J Cardiothorac Surg [24]	Japan, 08–16	36		VATS	LS-LND	11 (30.6)					80					
Nakagawa et al., 2020. Eur J Cardiothorac Surg [9]	Japan, 05–16	177	65.3 ± 8.9		SLND 123 (69.5) MLNS 54 (30.5)	0					57			34.9	27.5	
		160	60.9 ± 9.3		SLND 135 (84.4) MLNS 25 (15.6)	160 (100)					73.4			45.2	34.5	
Obiols et al., 2014. Ann Thorac Surg [14]	Spain, 04–10	26		Open + VATS	SLND	1 (3.9)				79	40					Excluded postoperative deaths
Yazgan et al., 2021. Acta Chir Belg [28]	Turkey, 07–20	69		Open + VATS	ns						24.0					
Yoon et al., 2019. Korean J Thorac Cardiovasc Surg [17]	South Korea, 05–14	22	64.1 ± 9.9		SLND						22.9					Patient underwent segmentectomy, (bi)lobectomy or pneumonectomy.

### 3.5. Disease-Free Survival Outcomes of uN2

Table 3 provides an overview of the characteristics of nine studies reporting on the long-term outcomes of patients with uN2 disease with complete resection, in which disease-free outcomes for 3y and 5y were reported in one paper. As depicted in the forest plot (Figure 6 and Figure 7), the pooled disease-free survival after 3y and 5y was 40% (95% CI 30–50%) (*n* = 134) and 31% (95% CI 24–38%) (*n* = 104).

In this paper, statistical improvement in disease-free survival after 3y and 5y was shown for patients with completely resected, uN2 disease treated with adjuvant therapy (*p* = 0.035) which is likely the cause of heterogeneity. The beneficial effect of adjuvant chemotherapy was also described by Kim et al. (*p* = 0.02) [10]. In the study by Kirmani et al. [22], a reduction in disease-free survival time (105 vs 127 months, *p* = 0.049) and a shorter time to recurrence (24.9 vs 68.2 months, *p* = 0.00065) were observed in patients with uN2 disease. Ghaly and colleagues [18] reported a trend for a decreased 5-year DFS in nodal upstaging (pN1-2) but this did not reach significance (*p* = 0.068). No further stratification for these factors was possible.

## 4. Discussion

Stage III NSCLC represents a heterogeneous group of patients ranging from resectable, early-stage disease to aggressive, locally advanced disease not suitable for surgical resection. Furthermore, the disease burden within mediastinal nodal involvement can vary from one lymph node metastasis to multiple nodes, skip or not, bulky or not, intracapsular or extracapsular involvement, and discrete or infiltrative, yet they are allocated to the same N descriptor. Therefore, it seems plausible that these characteristics result in a variance of prognoses and one specific treatment strategy does not fit all. Management of stage IIIA NSCLC remains a topic for discussion as more evidence emerges and no definite recommendations can be provided. Current guidelines on treatment of stage IIIA NSCLC demand case by case evaluation by an experienced multidisciplinary team determined by the feasibility to obtain an R0 resection and thus are highly influenced by the experience of thoracic surgeons [32]. Single modality therapy for N2 involvement does not yield rewarding long-term disease-free survival rates so multimodality regimens are currently administered in an act to improve life-expectancy. The role of surgery is still a matter of debate although randomized controlled trials have been performed [33]. However, in cases of uN2 disease the debate reappears after the intervention and focuses on the specific type of adjuvant therapy. Various studies have been demonstrating the survival benefit of surgery followed by postoperative chemotherapy in the management of patients with resectable N2 disease [34,35,36]. The assumption that surgery cannot be a valid treatment option for patients with N2 disease has been losing ground. As of yet, current guidelines only recommend surgery for proven single-station N2 disease, followed by adjuvant chemotherapy. 

Our results reveal a pooled prevalence of 7.97% with an OS rate of 44% of patients with uN2 disease who underwent surgical resection with curative intent. Substantial heterogeneity exists between the studies. We weren’t able to detect geographic differences in survival. Intercontinental differences in survival between patients with similar pN stage have been reported [31,37]. This is partially explained by the higher proportion of adenocarcinoma and EGFR-driver mutations in Asia and squamous cell lung cancer in Europe [38,39]. Additionally, a difference in retrieval and reporting of obtained lymph nodes between regions exists [39]. None of the included studies except for the study by Zirafa and colleagues [29] reported the use of Naruke lymph nodal map. In 2009, the International Association for the Study of Lung Cancer (IASLC) proposed a new lymph node map to gain more uniform data suitable for meaningful comparative analysis [40]. Prior to the introduction of the IASLC lymph node map, two maps were concurrently used by surgeons all over the world: the (Japanese) Naruke map and the Mountain-Dresler modification of the American Thoracic Society (ATS) map (MD-ATS). Discrepancy in the anatomical definition of specific lymph node stations in both maps made global data-analysis flawed. In particular, the interpretation of the subcarinal region and its relation to the bronchi. Analyses of the N descriptors in the IASLC international database revealed that subcarinal lymph nodes assigned to level 7 in the MD-ATS map (thus stage III) corresponded to either level 7 or 10 based on the anatomic borders defined in the Naruke map (stage III or II). Excluding the study by Zirafa [29] in the prevalence analysis did not solve heterogeneity.

A number of risk factors have been identified for discovering pN2 despite clinically staging as N0 or N1. In relation to tumor aggressiveness, a high SUVmax of the tumor on PET-scan [16,41,42,43] and adenocarcinoma [41,44,45] are recognized as independent risk factors. Centrally located large tumors and N1 involvement have a higher probability of invading the mediastinal lymph nodes, so consequently they are also associated with a higher risk for uN2. [16,41,46,47,48,49] However, these factors are accounted for in the staging work-up and adequate mediastinal staging according to the ESTS guidelines would have filtered out a majority of the patients at risk. However, in our study, strict adherence to the guidelines could not be verified. In one study by Yoon et al. [17] 77% of the patients with uN2 had centrally located tumors. If no suspected lymph nodes were identified by preoperative mediastinal imaging and the tumor was resectable, patients did not undergo additional invasive mediastinal staging. Due to improved staging and advancements in treatment modalities, we enter a novel era in which we seek the boundaries of different prognostic subgroups and their optimal treatment strategy. 

Reported prevalence rates of uN2 ranged from 2.8 to 25% depending on the implemented staging protocol [3,25]. In an attempt to minimize bias, we excluded studies that did not perform PET-scanning in their routine work-up and omitted invasive mediastinal staging. In 2007, the European Society of Thoracic Surgeons (ESTS) established a now widely used and validated algorithm on preoperative mediastinal staging that has since been revised to include PET-scan [2,50,51,52]. In 2005 Birim et al. [52] offered a comprehensive meta-analysis with proof of the added value of PET-imaging on mediastinal nodal assessment. FDG-PET imaging is to date the most accurate non-invasive diagnostic tool in our arsenal, with a sensitivity of 74–85% and specificity of 70–92% [52,53,54,55]. In contrast to CT scanning, an important shortcoming of a dedicated PET-scan is the limited ability for anatomical evaluation, but this will no longer be an issue when integrated PET-CT scanning will be more readily available. However, studies have reported overall rates of uN2 in PET-negative patients up to 15.3% upholding the need for further invasive mediastinal staging [56]. This can be partially explained by the reduced ability to detect carcinoids and adenocarcinoma on PET-imaging [9]. Results of a randomized ASTER-trial conducted by Annema et al. [57] in 2010, favored invasive mediastinal staging with both endosonography and mediastinoscopy. However, a 2019 meta-analysis by Bousema and colleagues [58] demonstrated similar rates of uN2 disease when staged with endosonography (EBUS or EBUS + EUS) regardless of confirmatory mediastinoscopy (9.6% vs 9.9%) and emphasized the higher risk for complications in performing the more invasive mediastinoscopy. A secondary observation that emerged from the ASTER-trial was that despite a significant difference in the uN2 rate (6.9% vs 14.3%) the 5-year survival remained exactly 35% [57]. We included two papers [21,26] in which endobronchial ultrasound-guided transbronchial needle aspiration (EBUS-TBNA) was the only method for nodal staging. However, these two papers did not yield similar rates, which can be explained by the need for experience and expertise for staging by endosonography. It can be challenging to collect a sufficient and representative sample of suspected lymph nodes. Some centers are even abandoning confirmatory mediastinoscopy as their experience in endosonography grew [27]. The results of the long-awaiting MEDIASTrial, a multicenter parallel randomized non-inferiority trial comparing the two diagnostic strategies led by Bousema as well, has been recently published and concluded that while accepting an uN2 rate of 8%, the more invasive mediastinoscopy can be omitted after negative systematic endosonography in patients with resectable NSCLC [59]. 

It is only speculation whether our results were affected by non-adherence to the guidelines for either preoperative or intraoperative staging, as representative lymph node assessment is the cornerstone for choosing the optimal treatment strategy and thus affecting survival outcomes. In theory, intraoperative lymph node retrieval has an influence on survival based on two beliefs. Firstly, more radical lymph node dissection leads to more accurate pathologic staging and thus assigning the patient to the most optimal treatment strategy. Secondly, the removal of metastatic lymph nodes effectively reduces tumor burden and would result in a true R0 resection, reducing the risk of recurrence. In our study we described three types of lymphadenectomies. A prospective randomized Z0030-trial did not show superiority of complete mediastinal lymph node dissection compared to sampling in terms of disease-free survival and OS outcomes but did report a rather modest incidence of uN2 (4%) [60]. Consistently, studies by Cerfolio [61] and Jeon [62] demonstrated higher uN2-rates when more radical lymph node dissection was performed. The study by Wang et al. [63] has shed more light on the potential benefit in survival with more extensive lymph node dissection. Mediastinal lymphadenectomy was identified as a significant prognostic factor for OS but dissection of ≥4 mediastinal nodes did not show an influential impact on long-term outcomes. Wang’s findings are supported by other authors, who saw the therapeutic role of more extensive lymph node dissection in the treatment of NSCLC when compared to lymph node sampling [64,65]. It is reasonable to believe that with more extensive lymph node retrieval, the risk of occult micrometastasis and skip metastasis can be eliminated and thereby reducing the risk for local recurrence and distant metastasis. Research into lobe-specific patterns of nodal metastasis has convinced some surgeons to accept that lobe-specific lymph node dissection (LS-LND) is sufficient [66,67,68,69]. However, different inclusion criteria and paucity of data on survival outcomes make it difficult to draw any definite conclusions about performing a less extensive lymph node dissection in the setting of uN2 disease. As mentioned in the Section 3, we weren’t able to examine the role of extent lymph node dissection on the prevalence and survival outcomes of uN2. However, we included one paper [21] that compared LS-LND with the golden standard of systematic mediastinal lymph node dissection (SLND). Kamigaichi and colleagues [21] reported a higher rate of uN2 when performing SLND which further emphasizes that a thorough assessment of the lymph nodes remains critical to ensure accurate staging in order to discover uN2 disease. 

The extent of nodal involvement ought to have a direct influence on long-term survival outcomes, as the number of involved lymph nodes, stations and the presence of skip metastases have been described as prognostic factors. We further sought the current evidence for these prognostic factors in uN2 disease. Anatomic studies suggest that metastatic spread evolves from one lymph node to multiple lymph node stations, thereby advancing disease burden [70]. However, the widely used TNM classification is a location-based categorization which fails to account for the extent of tumor burden at the level of the regional lymph nodes. Renaud and colleagues showed in two separate studies that microscopic N2 was associated with a better prognosis [71,72]. However, such characteristics of nodal metastasis are seldom reported, so comparative analysis is difficult to conduct. It seems logical that patients with microscopic levels of metastasis cannot be expected to have the same prognosis as those with multilevel involvement. Yet, regardless of the level of lymph node metastasis, they belong to the same category, N2. The possibility of nodal stratification based on the number of involved lymph nodes (nN) has been explored by few studies, since robust data are still lacking as the current staging methods are incapable of accurately counting involved lymph nodes [73,74,75,76]. Furthermore, systematic lymph node sampling can accurately determine the N-status but falls short in determining the quantitative level of N2 disease. Therefore, representative lymph node harvesting remains a hurdle in validating results from studies focusing on the number of lymph node metastases. More readily available are reports on single and multiple station involvement and its prognostic significance. Vast evidence demonstrates that survival outcomes are significantly better for patients with single station metastasis (34–39% 5y OS) compared to multiple station metastasis (11–22% 5y OS) in patients with NSCLC [64,77,78,79,80,81]. Several studies have reached the same conclusion for patients with uN2 disease (single 25–67% vs multiple 25–36% 5y OS) [82,83,84,85,86]. Although the 5-year survival outcomes (DFS and OS) of uN2 were statistically similar to those with pN1 in the study by Lee et al. [87], no statistical difference was found between patients with single versus multiple station metastases. Based on studies focused on skip metastasis, direct lymphatic drainage pathways appear to explain why the upper lobe is the predominant site of primary lung cancer linked to skip metastasis [88]. Based on this notion, some studies have explored the question whether lymph node stations 5 and 6 should not be considered N1 nodes for left upper lobe tumors, which would make pre-operative tissue confirmation for these cases less stringent [89,90,91]. A meta-analysis by Wang et al. [92] focusing on skip vs non-skip N2 in patients with pN2, reported higher 5-year survival rates in the skip-N2 group with an estimated odd ratio of 0.078 (95% CI 0.71–0.86, *p* < 0.001, I^2^ 67.1%). The prognostic impact of skip metastasis on uN2 hasn’t been established yet. Bille et al. [15] did not see a difference in median OS with or without skip metastasis (38.2 vs 37.5 months, *p* = 0.34). 

For the eighth edition of tumor, node and metastasis (TNM) classification, a new subdivision of nodal involvement was suggested based on the combination of location of involved lymph nodes, nN (single station vs multiple stations) and presence of skip metastasis: pN0, pN1a (single pN1), pN1b (multiple pN1), pN2a1 (single pN2 with skip metastasis), pN2a2 (single pN2 with hilar involvement), and pN2b (multiple N2) [32]. In accordance with this stratification, we haven’t been able to produce any relevant conclusions as data using this subdivision is still lacking. Furthermore, explorative analysis by Park et al. [30] and external validation conducted by Chiappetta et al. [93] have revealed that some of the prognostic implications of these subgroups overlapped. We expect more data to become available for the upcoming 9th edition of the TNM classification, to be published in 2024. Five of the included studies reported location of nodal involvement. In four studies, the subcarinal location was most frequently involved [15,16,17,30]. In the study by Amer et al. [20], the most frequent location of nodal metastasis was LN 2-4, secondly LN 7. This can be explained as the study included a majority of tumors located in the right upper lobe. Earliest findings of an associated worse prognosis with subcarinal disease in unsuspected uN2 dates back to 1996 with a reported 5-year survival rate of 22% [94]. In our study, two papers [21,24] that reported only lobe-specific lymph node dissection during their operations, did not explore the subcarinal region on a routine basis. Visser et al. [27] reported five cases with unreachable N2 metastasis (LN5-6-8) in their uN2-cohort which cannot be reached by cervical mediastinoscopy. Despite increased interest in sublobar resection in the treatment of primary lung cancer, the implications on long-term survival of patients with unforeseen mediastinal nodal disease are yet to be determined. A comparative study by Liou et al. [95] reported similar long-term survival outcomes in patients with occult N2 disease treated with either sublobar resection (46.6% 5y OS) or lobectomy (45.2%) for clinical stage Ia NSCLC. The authors emphasized the importance of adjuvant therapy and thus should not be delayed in favor of reoperation with complete lobectomy. It remains adamant that proper assessment and identification of hidden hilar and mediastinal nodal disease enables accurate staging, enhanced local control and, therefore, optimal adjuvant treatment and prognosis.

Studies have shown survival rates ranging from 34% up to 48% in patients with uN2 treated with adjuvant therapy which is comparable to survival rates of those with N1-disease treated with surgery primarily [8,14,87,96]. Although a study by Macia et al. [97] showed an inferior 5-year OS for single station uN2 of 25%, a similar pattern was observed. Patients with single uN2 had similar survival rates to that of multiple N1 (34%), in contrast to a 5-year OS of patients with single pN1 (73%). Several treatment strategies have been investigated but research into the role of adjuvant therapy on surgically treated patients has an inherent bias. In retrospective studies it is difficult to identify those who did not undergo adjuvant therapy due to surgical morbidity, but patients who receive surgical treatment presumable have better performance status and are therefore more likely to tolerate adjuvant therapy as well. Furthermore, published studies and trials generally include not only patients with pN2 but also those with pN0-1, so definitive conclusions need to be made cautiously. We included one study by Nakagawa et al. [9] that solely focused on the potential benefit of adjuvant therapy specific to patients with uN2 disease. They reported significant differences in long-term survival outcomes between patients with observation alone, and those treated with adjuvant chemotherapy. Nakagawa’s results are consistent with known trials favoring adjuvant chemotherapy in completely resected early-stage NSCLC with nodal involvement [98]. The long-awaited results of Lung ART-trial, an open-label randomized phase 3 trial, were published last year [99]. They studied the role of adjuvant radiotherapy in patients with completely resected pN2 NSCLC treated with (neo)adjuvant chemotherapy and found no significant difference in disease-free survival. However, patients with positive resection margins might still benefit from PORT (postoperative radiotherapy) for local control. The role of novel therapies in multimodality treatment strategies for early-stage NSCLC has yet to be determined but is expected to achieve an additional increase in overall survival. Systemic therapy targeted at epidermal growth factor receptor (EGFR) mutations for advanced-stage lung cancer, seems to produce satisfactory results in disease-free survival outcomes for patients with EGFR mutations [100]. Immunotherapy with anti-programmed death-1/programmed death ligand-1 agents (durvalumab) has proven its beneficial effect in patients with unresectable N2-disease and are expected to be evaluated in those with resectable N2-disease [101]. With the incoming era of novel targeted therapies and immunotherapy, accurate clinical and pathological nodal staging will become more important. In recent years, progress has been made in the study of molecular tumor markers but still lacks sufficient specificity and sensitivity to guide clinical treatment. Recently, Hao et al. [102] identified five potential biomarkers for detecting N2 lymph node metastasis in early-stage lung adenocarcinoma. Although further verification is lacking, identifying diagnostic biomarkers has great research prospects and can add more insight in preoperative nodal assessment. 

Our study had several limitations. All except for one paper were retrospective cohort studies with relatively low-quality evidence, therefore, the quality and validity of our meta-analysis was decreased. Various limitations may have influenced the interpretation of our results which have been elaborated in the Section 4. We included papers with a relatively small sample size and a substantial heterogeneity in the cohorts due to the vast amount of variables remaining present which did not allow further meaningful analysis. Noticing the heterogeneity in surgical procedures, ranging from sublobar resection to inclusion of pneumonectomy which is not a standard surgical procedure for cN0 NSCLC. Additionally, one study [14] reported survival rates after exclusion of postoperative deaths. As patients with uN2 disease are a specific population of patients who are inherently linked to an operation, it would be more truthful to include postoperative complications such as death. Lastly, we cannot account for possible inconsistencies of adherence to staging protocols of the involved institutes. Therefore, there may be patients who remain understated and the true prevalence may still be overestimated. Despite the major limitations and lack of definitive conclusions, we believe this review can guide future investigators in designing clinical trials focused on uN2 disease.

## 5. Conclusions

We report a rate of true uN2 disease of 7.97% despite adequate mediastinal staging. Whether surgery can play a pivotal role in N2 disease remains controversial. Attaining macroscopic and microscopic complete resection remains the cornerstone for surgical treatment and adjuvant therapy is applied in order to diminish the risk of locoregional recurrence and improve the OS rate. Yet, clear definitions of which subcategory of patients would benefit the most from such strategy are currently lacking. Moreover, the challenge remains not only to identify those at risk for uN2- involvement but also to include them in clinical trials that evaluate multimodality treatments. Awaiting results of ongoing trials, it seems patients with uN2 disease represent a subcategory with similar prognosis to stage IIb if complete surgical resection can be achieved, and the contribution of adjuvant therapy is to be further explored.

## Figures and Tables

**Figure 1 cancers-15-03475-f001:**
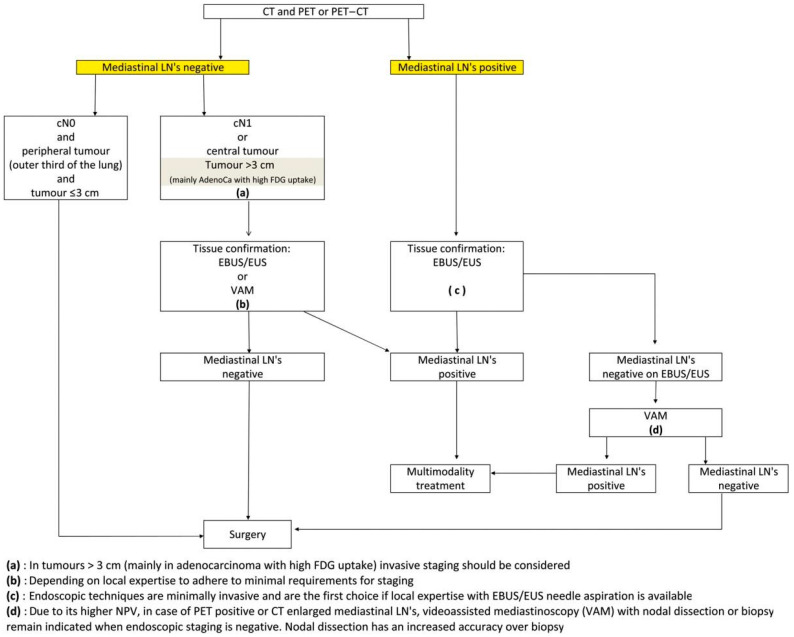
Revised ESTS guidelines for primary mediastinal staging (2014) [2].

**Figure 2 cancers-15-03475-f002:**
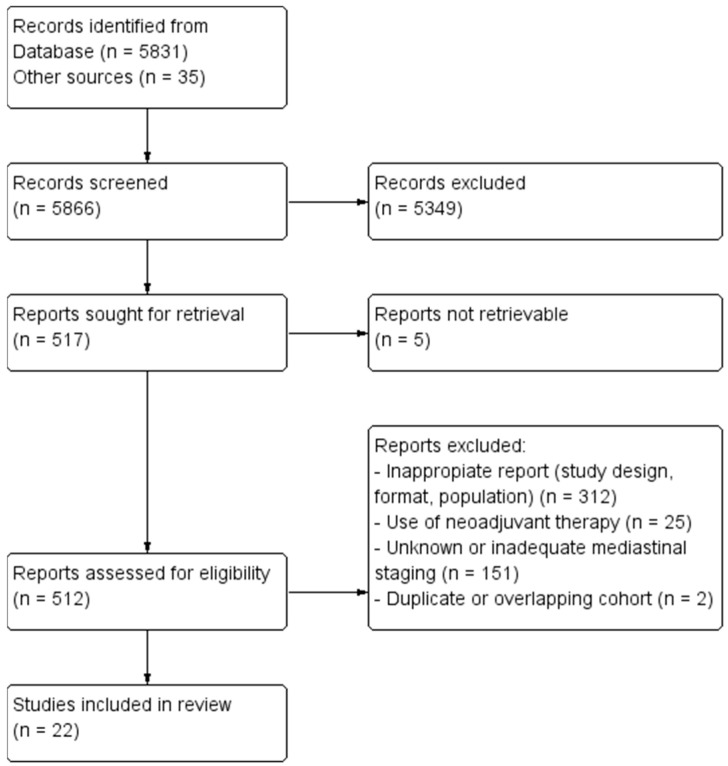
Study flow diagram according to PRISMA guidelines 2020 [7].

**Figure 3 cancers-15-03475-f003:**
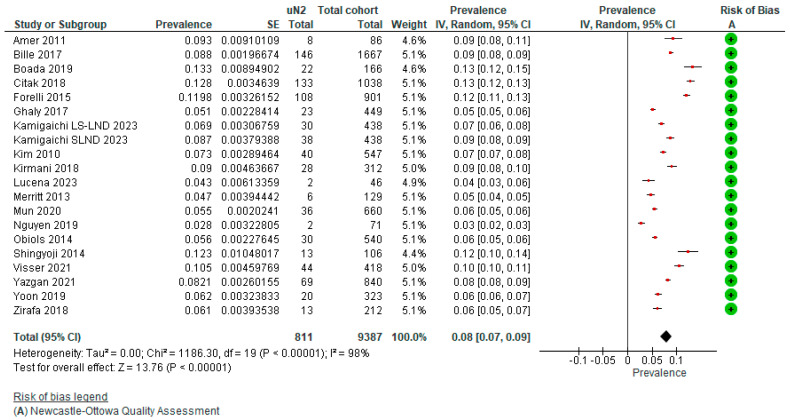
Pooled prevalence of uN2 disease + risk of bias assessment.

**Figure 4 cancers-15-03475-f004:**
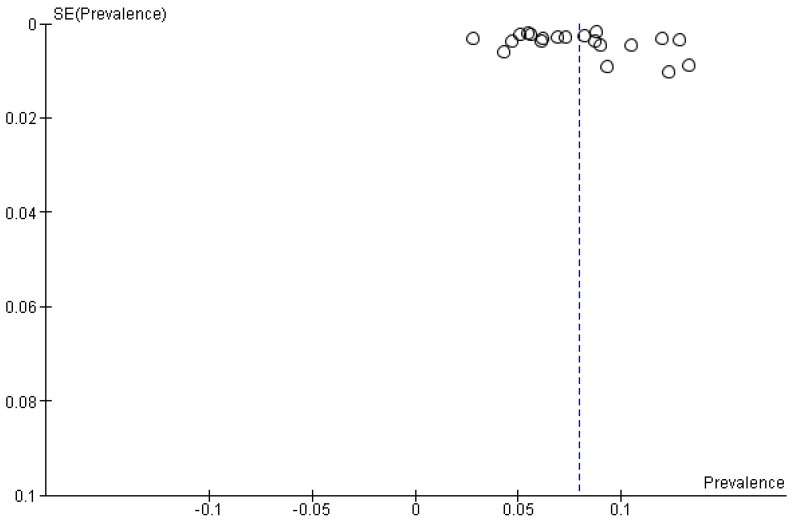
Funnel plot of pooled prevalence of the uN2 disease.

**Figure 5 cancers-15-03475-f005:**
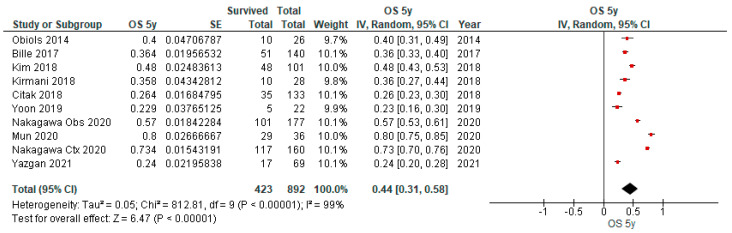
Forest plot of five-year overall survival of patients with uN2 disease and complete resection.

**Figure 6 cancers-15-03475-f006:**
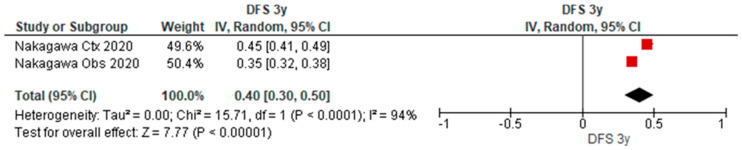
Forest plot of three-year disease-free survival of patients with uN2 disease and complete resection.

**Figure 7 cancers-15-03475-f007:**
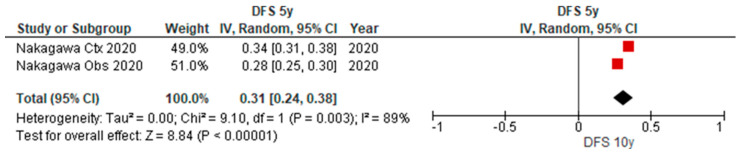
Forest plot of five-year disease-free survival of patients with uN2 disease and complete resection.

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
