# Peer review of "True Prevalence of Unforeseen N2 Disease in NSCLC: A Systematic Review + Meta-Analysis"

_cancers, 2023, doi:10.3390/cancers15133475_

Round 1

Reviewer 1 Report

The authors presented an analysis of 22 studies that identified patients with unanticipated N2 disease (uN2). The overall prevalence of true unanticipated pN2 (9387 patients) was shown to be 7.97% (95% CI 6.67–9.27%) with a pooled OS at five years (892 patients) of 44% (95% CI 31–58% ). The authors show that patients with uN2 disease represent a subcategory with a prognosis similar to stage IIb if complete surgical resection can be achieved and the contribution of adjuvant therapy is for further study.

The review is well structured, the material is correctly and consistently presented. I think that the article certainly deserves the attention of readers.

1. line 492, 501, 509 - typos? uN2N2

2. line 78 typo? cN0-1pN2 maybe cT0-1pN2?

Author Response

Dear reviewer 1,

I have resubmitted the revised version of manuscript with ID cancers-2402097, True Prevalance of Unforeseen N2 disease in NSCLC: a systematic review + meta-analysis.

Thank you for giving us the opportunity to revise and resubmit this manuscript. We appreciate the time and detail you have spent to evaluate the manuscript. We have incorporated all suggested changes by the editors and reviewers. The manuscript has certainly benefited from these insightful comments and suggestions. Changes to the manuscript are shown in red and my response to your comments are given below.

  1. line 492, 501, 509 - typos? uN2N2

All mention of "uN2N2" have been corrected to uN2 disease.

2. line 78 typo? cN0-1pN2 maybe cT0-1pN2?

I understand that this could have been seen as a typo but we deliberately refer to the specific group of patients who were preoperatively diagnosed with N0-1 (clinical stage cN0-1) but postoperatively as N2 and thus pathologically referred as pN2. We did not exclude any patients based on tumor size.

I hope the revised version is now suitable for publication and I look forward to hearing any further feedback.

Sincerely,

Hui Wing Kea
Corresponding author

Reviewer 2 Report

Very well written paper and very thorough presentation of the literature. My only issue is the very long discussion, which, in my eyes is not necessary and is difficult to follow. I would suggest shorting the discussion down to the necessary information. 

Author Response

Dear reviewer 2,

I have resubmitted the revised version of manuscript with ID cancers-2402097, True Prevalance of Unforeseen N2 disease in NSCLC: a systematic review + meta-analysis.

Thank you for giving us the opportunity to revise and resubmit this manuscript. We appreciate the time and detail you have spent to evaluate the manuscript. We have incorporated all suggested changes by the reviewers and editors. The manuscript has certainly benefited from these insightful comments and suggestions. Changes to the manuscript are shown in red. We have shortened the discussion to be more concise for the readers.

I hope the revised version is now suitable for publication and I look forward to hearing any further feedback.

Sincerely,

Hui Wing Kea
Corresponding author

Reviewer 3 Report

The study is very complex and articulated and focuses on a topic of wide debate in the literature.

I recommend focusing more precisely on the objective of the review, especially in the abstract.

Table 1- Study of Zirafa and colleagues: the origin of the study is Italy instead of France

Line 282. The authors should explain why they consider the subcarinal station as level 7 or 10 in the naruke map.

In consideration of the different classifications of the lymph node stations proposed by the various studies, a comparative table of LN stations could be useful.

I only suggeste a more concise discussion

Author Response

Dear reviewer 3,

I have resubmitted the revised version of manuscript with ID cancers-2402097, True Prevalance of Unforeseen N2 disease in NSCLC: a systematic review + meta-analysis.

Thank you for giving us the opportunity to revise and resubmit this manuscript. We appreciate the time and detail you have spent to evaluate the manuscript. We have incorporated all suggested changes by the reviewers and editors. The manuscript has certainly benefited from your insightful comments and suggestions. Changes to the manuscript are shown in red and my response to your comments are given below. We have shortened the discussion to be more concise for the readers.

  • Table 1- Study of Zirafa and colleagues: the origin of the study is Italy instead of France

I have corrected this and it did not impact our results or conclusions made in the manuscript.

  • Line 282. The authors should explain why they consider the subcarinal station as level 7 or 10 in the naruke map.

I have rephrased this section to emphasize the findings of the 2009 paper by the IASLC Lung Cancer Staging Project-group. Due to differences in the anatomic definitions for each lymph node station, subcarinal lymph nodes that were labelled as level 7 in the MD-ATS map could have been appointed as levels 7 and 10 according to the Naruke map as the borders of the lymph node stations aren’t as clearly defined. As a result, some tumors staged as N2, stage IIIA according to the MD-ATS map, were staged as N1, stage II by the Naruke map. In the IASLC map, subcarinale lymph nodes are level 7, N2 nodes.

  • In consideration of the different classifications of the lymph node stations proposed by the various studies, a comparative table of LN stations could be useful.

As all included studies, except for the study by Zirafa used the same lymph node map, a comparative table would definitely be interesting but in our opinion not crucial for this manuscript. Therefore, with the aim to keep the manuscript focused and concise, we would recommend readers to read the paper by IASLC Lung Cancer Staging Project-group published in 2009 (reference 41) in which the different classifications are extensively discussed.

I hope the revised version is now suitable for publication and I look forward to hearing any further feedback.

Sincerely,

Hui Wing Kea
Corresponding author

Reviewer 4 Report

no negative comments

extensive and very cohmprensive review

almost 10k patients included in the analysys

estesive literature review with more than 500papers screened

in my opinion the manuscript deserves acceptation outright

Author Response

Dear reviewer 4,

I have resubmitted the revised version of manuscript with ID cancers-2402097, True Prevalance of Unforeseen N2 disease in NSCLC: a systematic review + meta-analysis.

Thank you for giving us the opportunity to revise and resubmit this manuscript. We appreciate the time and detail you have spent to evaluate the manuscript. We have incorporated all suggested changes by the reviewers and editors. The manuscript has certainly benefited from these insightful comments and suggestions. Changes to the manuscript are shown in red and I'd like to thank you again for your kind words.

I hope the revised version is now suitable for publication and I look forward to hearing any further feedback.

Sincerely,

Hui Wing Kea
Corresponding author